# Advancing the CMS Level-1 Trigger: Jet Tagging with Deep Sets at the HL-LHC

Stella F. Schaefer[1,2]*, Christopher E. Brown[1], Duc M. Hoang[1,3], Sioni P. Summers[1], Sebastian Wuchterl[1] on behalf of the CMS Collaboration. All authors contributed equally to this work.

**1** CERN (Geneva, Switzerland) **2** University of Hamburg (Hamburg, Germany)
**3** Massachusetts Institute of Technology (Cambridge, USA)
* stella.felice.schaefer@cern.ch

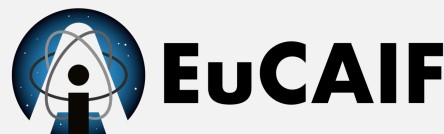

*The 2nd European AI for Fundamental Physics Conference (EuCAIFCon2025) Cagliari, Sardinia, 16-20 June 2025*

## Abstract

At the High Luminosity LHC, selecting important physics processes such as (di-) Higgs production will be a high priority. The Phase-2 Upgrade of the CMS Level-1 Trigger will reconstruct particle candidates and use pileup mitigation for the 200 simultaneous proton-proton interactions. A fast cone algorithm will reconstruct jets from these particles, providing access to jet constituents for the first time. We introduce a new multi-class jet tagger with a small, quantized Deep Sets neural network. The tagger, trained on a mix of simulated CMS events, predicts various hadronic and leptonic classes. We present the tagger, its performance, and its improvements for triggering on (di-) Higgs events.

## 1 Introduction

The High Luminosity upgrade of the LHC at CERN will bring new opportunities to study open questions of the standard model. Its increased luminosity will lead to a higher absolute number of rare physics processes. Especially, di-Higgs physics will be one of the major research areas. The data-taking conditions at High Luminosity will be more challenging than before, requiring new tools and strategies for effective trigger selection. One of the most important updates is the addition of tracking information to the CMS Level-1 Trigger (L1T) [1]. It allows for the inclusion of the Particle Flow (PF) algorithm [2], combining information from multiple subsystems to infer the particle identities. Furthermore, the pileup per particle identification (PUPPI) algorithm provides reliable pileup mitigation [3]. Going beyond the addition of established algorithms like PF and PUPPI, tracking at L1T also enables us to study the individual jet constituents, thus allowing for efficient jet flavor tagging. It adds to the limited description of jets and allows for more targeted trigger selections, going beyond pure kinematic selections. This helps selecting physics processes, like di-Higgs production, through tagging of the decay

products. Jet tagging will be implemented at the the second layer of the correlator trigger subsystem, making use of the reconstructed Seeded Cone (SC) jets [4] and providing jet flavor information to seeds at the global trigger. This work unifies the tagging categories b, light, charm, gluon jets as well as hadronically decaying tau leptons, electron and muons in one multiclass model. Additionally, the tau leptons category is split into two for positively and negatively charged taus, amounting to a total of 8 categories. Previous tagging developments have only focused on b-jets and hadronically decaying tau leptons ($\tau_h$) [5,6].

## 2  Tagging strategy

To effectively employ jet tagging at L1T, jet constituents are crucial. The jet constituents are clustered to form jets using the SC algorithm. It is specifically tailored to run on FPGAs at L1T and matches the performance of standard jet reconstruction algorithms used outside the CMS L1T. Specifically, the model is trained using the 16 leading transverse momentum ($p_T$) constituents of SC jets. Each constituent is described by a mixture of kinematic features, Particle IDs (PID) from PF, as well as a "record filled" feature, which is one for real constituents and zero for padded constituents. Zero-padding of constituents is used to create a regular input shape, necessary for the application at L1T. To train the jet tagger, a total of around 21 million jets from multiple Monte Carlo simulated physics processes are included, from which 10% are reserved for testing and 9% for validation. The classes are roughly balanced in the dataset, but class weights are used to ensure equal importance during training. Code for all steps from training to evaluation can be found on GitHub.[1]

### 2.1  Network architecture

A jet tagging network must fit within the strict hardware and latency constraints of the L1T, thus the best performing model is not necessarily the best choice. The latency and resource consumption of state of the art architectures, like ParT [7], are incompatible with the constraints at L1T. However, a small, quantized Deep Sets (DS) model, inspired by Ref. [8], meets all requirements. Additionally, this architecture is permutation-invariant, making sorting of constituents only necessary to identify the leading $p_T$ constituents. Figure 1 shows the model architecture in detail. The model is also tasked to regress the $p_T$ of the jet from the constituents in a separate branch, going beyond jet classification [9].

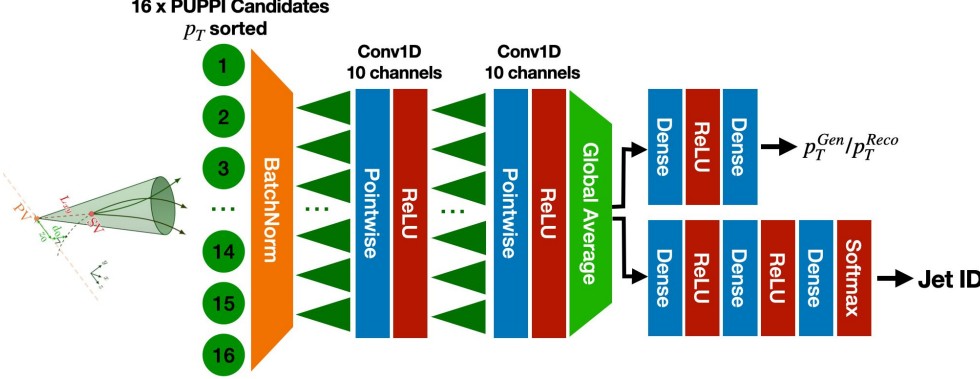

Figure 1: Deep Sets architecture for jet tagging and $p_T$ regression. All 16 constituents are processed individually before aggregating them through a global average and splitting into a $p_T$ regression branch and a jet ID branch [9].

---

[1] https://github.com/CMS-L1T-Jet-Tagging/TrainTagger

# 3   Jet tagging performance

To evaluate the performance of the jet tagger, one-vs-one receiver operating characteristic (ROC) and area under curve (AUC) scores of particular interest are presented in Figure 2. These are evaluated on the $t\bar{t}$ process. The process, describing the production of a pair of top quarks and their subsequent decay, can contain all of the tagged flavors in the final state through the different decay modes of the top quark. A light-vs-gluon score could help identify characteristic jets from vector boson fusion (VBF), an important (di-) Higgs production process, whereas a b- vs light-jet or $\tau_h$ vs. all-jets score could help separate b-jets and $\tau_h$-jets from light-jets. Generally, the model performs best for the charged lepton classes, specifically electrons and muons. These classes gain most from the PID input features. The other classes exhibit worse performance, likely due to confusion between the b- and charm-jet classes as well as the light- and gluon-jet classes.

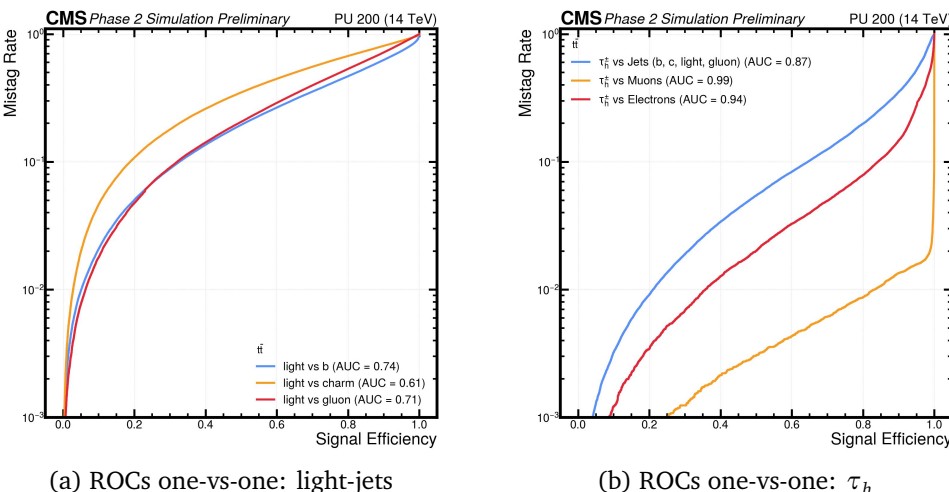

(a) ROCs one-vs-one: light-jets        (b) ROCs one-vs-one: $\tau_h$

Figure 2: One-vs-one ROC curves and AUC scores evaluated on a $t\bar{t}$ physics sample [9].

# 4   Seed studies

The best way to demonstrate the benefits of jet tagging at L1T is by developing dedicated trigger selection paths, called seeds. Without jet tagging, seeds often rely purely on kinematic information to tag processes like di-Higgs. One such seed requires the sum of all jet $p_T$, referred to as $H_T$, to exceed 370 GeV and a minimum of four jets, resulting in a rate of 14 kHz, similar to previous studies [1]. A smaller $H_T$ threshold at 220 GeV can be achieved through the inclusion of b-tagging information for the 4b final state information. To guarantee a fair comparison between these seeds, the threshold applied to the sum of the b-tags of the leading $p_T$ jets is chosen such that the seed results in the same rate of 14 kHz. The comparison is shown in Figure 3a. For these, all selected events are binned against the generator $H_T$, since this distribution is sensitive to the Higgs self coupling $\kappa_\lambda$. Although the absolute efficiency $\int \epsilon$ is similar, there is significant efficiency gain at low $H_T$. This is especially important, since large fractions of events in this region are lost through the $H_T$-only seed. However, the tagging seed does not reach the same plateau efficiency due to limitations of the tagging capabilities. Ultimately, a seed using a logical or between both seeds allows for optimal tagging of di-Higgs $\rightarrow$ 4b.

A similar comparison is shown in Figure 3b. It compares the multiclass jet tagger to the

previous b-tagging developments made for Phase-2. The multiclass jet tagger outperforms by 30% in total efficiency, showing improvements in the complete $H_T$ spectrum.

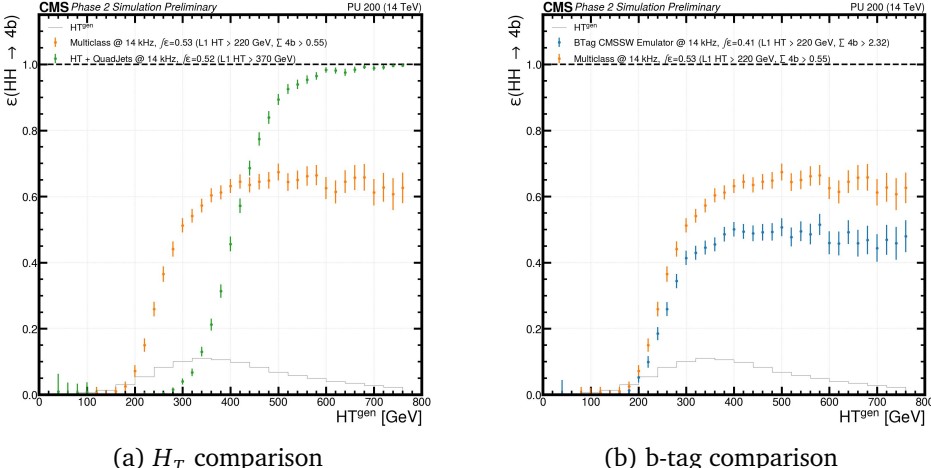

(a) $H_T$ comparison         (b) b-tag comparison

Figure 3: Comparison of a purely kinematic $H_T$ seed and tagging + $H_T$ as well as a comparison to previous b-jet tagging developments for the HH → 4b process. The new seed shows significant efficiency gain at low $H_T$ [9].

## 5   FPGA implementation

The complete jet reconstruction including jet tagger has been implemented for a prototype Phase 2 L1T hardware – a Serenity ATCA blade with AMD VU13P FPGA [10]. The total latency is 1.011 µs, of which 234 ns are tagging. This is slightly in excess of the target 1 µs of the correlator trigger second layer, but further optimizations are expected to bring the latency within constraints. The tagger throughput is sufficient to tag all reconstructed jets, at the clock rate of 360 MHz. The tagger and its input preparation utilize 13% of device LUTs, 8% of FFs, 14% of DSPs, and 0.6% of BRAMs, fitting well within one FPGA. A comparison to the CMS Software Simulations for simulated physics samples also matches exactly with the model's performance on an FPGA, showing that the performance presented here will be attainable in the hardware system.

## 6   Conclusion

Jet tagging will be a crucial addition to the Phase-2 CMS Level-1 Trigger (L1T) to face the challenges brought by High-Luminosity Upgrade of the CERN LHC. It extends the jet description beyond kinematic properties, benefiting multiple seeds in the global trigger. A multiclass Deep Sets model presents a viable option for jet tagging at L1T. It classifies into uds, c, b, gluon, $\tau_h^+$, $\tau_h^-$ as well as electron and muon, going far beyond previous developments. The leptons, especially electrons and muons are the most reliably predicted classes, whereas the charm class performs worst, probably due to confusion with other classes. Seed studies for HH → 4b show great improvements through the inclusion of jet tagging information. A comparison to an $H_T$-only seed shows similar absolute efficiency, but a significant increase at low $H_T$, advocating a logical or of both seeds. The complete jet tagger and input preparation fits well onto one VU13P-2 FPGA and fulfills the latency requirements, while still allowing some improvements and additions to the model.

## Acknowledgements

This work has been [partially] funded by the Eric & Wendy Schmidt Fund for Strategic Innovation through the CERN Next Generation Triggers project under grant agreement number SIF-2023-004.

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
