# Peer review of "Advancing the CMS Level-1 Trigger: Jet Tagging with DeepSets at the HL-LHC"

_SciPost Physics Proceedings_

## Round 1 · Referee Report · Anonymous (Referee 1) · 2025-11-10

Strengths

Further deployment of processing algorithms on FPGAs, turning these into online
processing, is always appreciated and valuable.

Weaknesses

  • Model design procedure is not detailed.
  • FPGA deployment/synthesis workflow is not provided.
  • The data itself is not described.
  • The above weaknesses are most likely resulting from the page constraints. However, there is enough room to improve the data description.

Report

The topic and content are very relevant. My remarks are given in the "Requested changes" segment.

Requested changes

1- Introduction: Capitalise "particle flow (PF)" -> "Particle Flow (PF)" 2- All sections and subsection: These are not titles, thus only the first word is to be capitalised. 3- Section 2: You have mentioned that 10% of the data is reserved for testing and validation. What are the portions, individually? As testing and validation sets cannot be mixed. If the considered test set is rather small, it will not mitigate the risk of overfitting. 4- Section 2: Regarding the padding, is it constant or variable? 5- Section 2.1: "hardware and latency constraints" What are these constraints? Provide numbers. 6- Section 2.1: "DeepSet" is written as two separate words in the rest of the paper. 7- Section 3: "Fig. 2a" You have abbreviated figure term for sub-figures. Needs to be adjusted in LaTeX packages. 8- Section 3, paragraph 2: A new paragraph has to start indented. 9- Section 3: The class balance statistics of the dataset is not provided. How many classes and what are the proportions? 10- Generally speaking, plots are too small to be legible. 11- Section 5: Provide a reference or a footnote to the FPGA's specification page/document from the manufacturer. 12- Section 5: "models" -> "model's" 13- A code repository is linked to the SciPost submission. This should be added to the paper either as a reference (preferred if the link is permanent), or a footnote.

Recommendation

Ask for minor revision

---

## Round 2 · List of Changes

Changes made for this resubmission:
Remove ROC OvR and 4b mHH performance showcase plots and slight changes to the texts that reference these plots
Include more detailed information on deployment on FPGA + reference to FPGA specifications
add information on training, validation, testing splitting
Fig. abbreviation -> Figure
add footnote containing GitHub reference

---

## Editorial Decision

accepted_in_target_journal